# Machine Learning to Predict Pregnancy in Dairy Cows: An Approach Integrating Automated Activity Monitoring and On-Farm Data

**DOI:** 10.3390/ani14111567

**Published:** 2024-05-25

**Authors:** Thaisa Campos Marques, Letícia Ribeiro Marques, Patrick Bezerra Fernandes, Fabio Soares de Lima, Tiago do Prado Paim, Karen Martins Leão

**Affiliations:** 1Departamento de Zootecnia, Instituto Federal Goiano, Rio Verde 75901-970, Brazil; tcmarques@ucdavis.edu (T.C.M.); leticiamarqueszootec@gmail.com (L.R.M.); bezerrazpatrick@gmail.com (P.B.F.); tiago.paim@ifgoiano.edu.br (T.d.P.P.); 2Department of Population Health and Reproduction, University of California, Davis, CA 95616, USA; falima@ucdavis.edu

**Keywords:** fertility, reproduction, dairy cattle, statistical inferences

## Abstract

**Simple Summary:**

Scientists have developed a way to more accurately predict when dairy cows are most likely to become pregnant using automated activity monitoring (AAM) systems to track their activity. These systems track the cow’s movement and behavior in real time, which is crucial for determining the best time for artificial insemination (AI). This study used data from over a thousand Holstein cows to create a mathematical model that predicts pregnancy chances at the time of AI, considering not just the cow’s activity data but also individual health, the environment, and even the specific bull used for insemination. This study found that combining on-farm data (like health and environmental conditions) with the AAM data gives a clearer picture of a cow’s pregnancy chances compared to using AAM data alone. The random forest model, one of the mathematical methods used, was particularly good at reducing errors in prediction. This research suggests that merging detailed farm data with automated monitoring can greatly improve the predictions of pregnancy at the time of AI, which is beneficial for managing dairy cow reproduction efficiently.

**Abstract:**

Automated activity monitoring (AAM) systems are critical in the dairy industry for detecting estrus and optimizing the timing of artificial insemination (AI), thus enhancing pregnancy success rates in cows. This study developed a predictive model to improve pregnancy success by integrating AAM data with cow-specific and environmental factors. Utilizing data from 1,054 cows, this study compared the pregnancy outcomes between two AI timings—8 or 10 h post-AAM alarm. Variables such as age, parity, body condition, locomotion, and vaginal discharge scores, peripartum diseases, the breeding program, the bull used for AI, milk production at the time of AI, and environmental conditions (season, relative humidity, and temperature–humidity index) were considered alongside the AAM data on rumination, activity, and estrus intensity. Six predictive models were assessed to determine their efficacy in predicting pregnancy success: logistic regression, Bagged AdaBoost algorithm, linear discriminant, random forest, support vector machine, and Bagged Classification Tree. Integrating the on-farm data with AAM significantly enhanced the pregnancy prediction accuracy at AI compared to using AAM data alone. The random forest models showed a superior performance, with the highest Kappa statistic and lowest false positive rates. The linear discriminant and logistic regression models demonstrated the best accuracy, minimal false negatives, and the highest area under the curve. These findings suggest that combining on-farm and AAM data can significantly improve reproductive management in the dairy industry.

## 1. Introduction

Reproductive performance is key to the profitability of dairy farms. Several factors, encompassing cow health, management, and environmental conditions, can influence whether a cow becomes pregnant after artificial insemination. These include the body condition score [1], retained placenta and periparturient diseases [2], parity [3], nutrition, heat stress, the month of artificial insemination [4], and environmental factors [5]. In this context, effective estrus detection is a determining factor in reproductive performance in intensive dairy production systems, as these cows have a lower peak of activity and estrus duration [6,7]. Thus, the use of the Automated Animal Monitoring (AAM) systems provides improved reproductive performance through accurate estrus detection, consecutive increases in service rates [8], and a reduction in the days to the first AI [9]. Considering the increased animal activity, AAM detects 15 to 35% more cows in estrus compared to visual observation, achieving an efficient detection rate of over 80% [10]. Nevertheless, our group observed that the inclusion of cow-specific variables and environmental factors in the statistical model can help increase more precisely estrus detection in dairy cows [11]. In addition, the AAM available commercially, while effective in detecting estrus, nowadays, does not provide pregnancy probability estimation, which could enhance semen investment management. Integrating statistical models to predict pregnancy probabilities would allow for more strategic use of high-value semen in cows with higher chances of conception, thereby optimizing genetic gains and improving returns on investment.

Statistical models have been used to predict lifetime production [12], fertility [13,14], health [15], and genomic selection [16]. Machine learning models are an alternative approach to classical statistical models for developing predictive models in large datasets, such as livestock-related studies [17], allowing for proactive management decisions and the customization of approaches to suit specific farm conditions [18]. In cows, studies have been conducted using this powerful tool to predict pregnancy success [19], and health disorders such as clinical mastitis, subclinical ketosis, lameness, and metritis [20].

In summary, while it is evident that the use of AAM can improve estrus detection, studies including cow-level variables and environmental factors associated with fertility in AAM machine learning models are still lacking. We hypothesized that incorporating on-farm factors like parity, peripartum health history, and environmental conditions in AAM models to predict pregnancy improves the models’ accuracy, sensitivity, and specificity using different machine learning algorithms. Therefore, the objective of this study was to compare the effect of including cow-level variables and environmental factors associated with fertility in predicting pregnancy at the time of AI in dairy cows using AAM through machine learning models.

## 2. Materials and Methods

### 2.1. Data and Animals

Our dataset was from a previous study comparing two artificial insemination (AI) times after the AAM alarm to predict pregnancy in dairy cows [11], where the methodology is thoroughly detailed. The retrospective observational case–control study briefly used data collected from a commercial dairy farm in the southwestern state of Goiás, Brazil. These data, collected using the farm’s software, covered the period from January 2018 to December 2020. Briefly, the study utilized data from 1054 Holstein cows, producing an average of 11,154 kg of milk per animal per lactation (305 days). These cows were housed in a free-stall barn equipped with fans above the stalls and sprinklers along the feedline, and they were milked three times daily. Data regarding cow-level [age, parity, body condition score (BCS, 1 = thin and 5 = fat, according to Ferguson et al. [21]), and locomotion score (LS, 1 = walked normally and 5 = presented lameness, according to Sprecher et al. [22]), days in milk (DIM), milk production, somatic cell count (SCC), and retained fetal membranes (RFMs)], peripartum diseases (hypocalcemia, ketosis, displacement abomasum, laminitis, foot disorders, pneumonia, and clinical mastitis) were recorded by farm personnel. The Metricheck^®^ device (SimcroTech, Hamilton, New Zealand) was used to collect vaginal discharge at 11 ± 4 days postpartum to be classified according to Sheldon et al. [23]. Cows with more than 50% pus or fetid watery reddish-brown discharge were defined with metritis and treated in a single dose with 20 mg ceftiofur hydrochloride (Lactofur^®^, Ouro Fino, Cravinhos, Brazil).

The daily time (in minutes) of activity and rumination was measured using AAM (SCR^®^, Netanya, Israel), which generated the estrus intensity to predict fertility on a scale from 0 to 100, where 0 represents no fertility and 100 represents high fertility.

Cows were synchronized for the first AI using prostaglandin (PGF 0.5 mg cloprostenol), or an estradiol- and progesterone-based program (EP) as described by Pereira et al. [24]. On Day 0, cows received two progesterone intravaginal implants (CIDR devices, Zoetis, Florham Park, NJ) and 2 mg of estradiol benzoate. After 7 days (Day 7), cows were treated with PGF (0.5 mg cloprostenol). On Day 9, they received 1.0 mg of estradiol cypionate, and the CIDR devices were removed. The AI was performed at 60 ± 7 DIM in one of two different time periods, at 8 h (*n* = 536) or 10 h (*n* = 518) after the AAM alarm with estrus intensity greater than 30. The pregnancy diagnosis was performed by the veterinarian responsible for overseeing the farm 30 days after AI.

The seasons followed for the Southern Hemisphere: spring (21 September to 20 December), summer (21 December to 20 March), fall (21 March to 20 June), and winter (21 June to 20 September). Environmental temperature and relative humidity (RH) were obtained from the weather station situated on the farm (ADAMA Clima^®^, Adama Brasil, Londrina, Brazil). To assess the level of heat stress the animals experienced, the temperature and humidity index (THI) was calculated on the day of the AI using the model defined by Mader et al. [25]: THI = 0.8 × T + [(RH (%) ÷ 100) × (T − 14.4)] + 46.4, where T represents the ambient temperature in °C, and RH stands for relative humidity.

### 2.2. Building Prediction Models

First of all, the non-continuous variables described in this study above were categorized for each cow included in the research: parity (1 = primiparous, 2 = multiparous), RFM (yes vs. no), BCS (1 = thin to 5 = fat), LS (1 = normal, 2 = mild lameness, 3 = moderate lameness, 4 = lameness, 5 = severely lameness), peripartum diseases (yes vs. no), season of the year at AI (spring, summer, fall, and winter), vaginal discharge at 11 ± 4 d postpartum (1 = clear or translucent, 2 = little purulent material, 3 = mucopurulent, 4 = 50% or more pus, 5 = fetid watery reddish-brown), reproductive program (PGF vs. EP), AI time (8 h vs. 10 h), bull (1 to 10), and pregnant (yes vs. no).

Secondly, we selected six models based on maximizing the Jaccard dissimilarity between sets of models from the classification models available on the package caret [26] in R software v.4.2.2 [27]. Therefore, the models selected included logistic regression (L), random forest (RF), linear discriminant analysis (LDA), support vector machines with linear kernel (SVM), Bagged AdaBoost (ADABAG), and Bagged Classification Tree (TREEBAG).

The collinearity of the variables was verified and pairs of variables with VIF higher than 10 were removed. This step removed the air temperature measures as it had higher VIF with the temperature and humidity index (THI). Then, we evaluated the models with five different sets of variables (FULL, AAM, no-AAM, GA, and STEP), as explained next (Table 1).

We evaluated the models using all variables available (FULL), only the three variables (activity, rumination, and estrus score) provided by the monitoring system (AAM), and all other variables available without those from the monitoring system (no-AAM). The GA group of variables was selected by the method of genetic algorithms using the function gafs of the caret package. This method is recognized for its effectiveness in feature selection, a crucial step in data mining, removing irrelevant and/or redundant features from a dataset [28].At the end, we applied stepwise feature selection method in logistic regression and linear discriminant regression to select the best set of variables for building the prediction model based on Akaike Information Criterion (AIC). Then, we had 6 methods with 4 sets of variables, resulting in 24 different models. Moreover, we have L and LDA methods with the STEP set of variables, ending at 26 different models to be evaluated.

The dataset was split into training (80% of the observations) and test set (20% of the observations). The predictors were centered and scaled using the preProcess function of the caret package. Then, the 26 different models were run using the train function of the caret package, setting family equal to binomial, as the outcome was pregnancy status (yes or no). Three repeats of 10-fold cross-validation were performed on the training data. For the model using treebag method, the nbagg was set as 50 and the metric used was based on the area under the curve (AUC). Random forest method also used the AUC metric for optimization.

Accuracy, false positive rate, and false negative rate were calculated building a confusion matrix of the predicted by the model and the observed in the test data. The area under the curve of each model was calculated using the predict function. The statistics of goodness of fit were reported as sensitivity (Se), specificity (Sp), positive predictive value (PPV), and negative predictive value (NPV).

## 3. Results

The evaluation of various machine learning models to predict pregnancy in high-milk-producing cows revealed distinct performance metrics across the models. The specificity of most of the models exceeded 0.80, except for those utilizing the no_AAM variables. The sensitivity exceeded 0.60 in only eight of the FULL, STEP, and GA models. Additionally, the accuracy surpassed 80% in nine models, predominantly in the FULL, STEP, and GA setups. The highest AUC values were observed in the GA (logistic) and FULL (rf) setups, achieving up to 0.87. These results underscore the importance of including cow-level and environmental variables to enhance the models’ predictive accuracy.

We evaluated the performance of each machine learning algorithm according to its accuracy, sensitivity, specificity, PPV, NPV, FP, FN, and AUC value (Table 2).

The SVM model using the no_AAM variables did not converge properly. The specificity of the models exceeded 0.80, except for no_AAM (AdaBag and treebag). By contrast, in the sensitivity of 26 models, only 8 were greater than 0.60, being from the FULL (lda, logistic, and svm), STEP (logistic), and GA (Ida, logistic, svm, and treebag) setups. The accuracy of only nine models was higher than 80%, namely, FULL (lda, logistic, rf, and svm), STEP (logistic), and GA (Ida, logistic, svm, and AdaBag). The AUC reached 0.87 in GA (logistic) and 0.86 in FULL (rf), STEP (logistic), and GA (Ida, svm, and rf). Using only the no_AAM data did not perform well in any of the best goodness of fit measures. The results of the machine learning algorithms models to predict pregnancy at the first postpartum AI in high-milk-producing cows using AAM showed that the inclusion of the cow-level variables and environmental factors associated with fertility improves the prediction of pregnancy success.

Most of the models using FULL, STEP, and GA increased the accuracy, sensitivity, and specificity in predicting pregnancy compared to the models that did not use AAM or only used variables from AAM. The positive and negative predictive values were higher in most of the models compared to the no_AAM model. All these values are reflected in a higher area under the curve compared to the no_AAM model. Moreover, the inclusion of the on-farm variables in the AAM model is 3–4% more accurate in predicting pregnancy compared to using just AAM parameters.

The analysis of machine learning showed that the addition of farm-available parameters, particularly parturition-related disorders, season on AI day, and maximum THI on AI day to the AAM-related variables models was necessary to predict whether the cow would become pregnant or not with an accuracy of 84–85% (Figure 1). Moreover, the different measures demonstrated the best models depending on the use of them. In terms of positive pregnancy prediction, the random forest model with all the data (FULL) provides the best results. On the contrary, in terms of equilibrium between the accuracy, AUC, and negative predictive value, the logistic (STEP and GA), support vector machines (GA), and linear discriminant (GA) methods were the best options. Thus, this preliminary study demonstrated the potential benefits of using machine learning models for predictive classification.

## 4. Discussion

The best models in terms of the AUC were based on the logistic regression, linear discriminant, and support vector machine methods. The group of variables used in these models (GA) was composed of the AAM data (activity at estrus day and estrus score) coupled with the cow health data (retained fetal membranes and vaginal discharge score) and the environmental variables (season and THI max of the day of AI). Therefore, the inclusion of only these four variables resulted in a better prediction of pregnancy probability. It is worth noting that, currently, without any additional information, the expected probability is 50% for all cows [29]. In addition, Hansen [30] mentioned that the application of machine learning to identify morphological features could be enhanced by incorporating information about pregnancy outcomes after embryo transfer.

The current study highlights the importance of continuously monitoring and recording cow health data. Monitoring the retained fetal membranes is just a matter of keeping a record of animals identified with this disorder, which causes a decline in fertility and milk output, leading to economic losses [31,32]. Furthermore, it necessitates additional mating attempts, prolongs the interval between calvings, and may lead to health complications, including the slow recovery of the uterus, ovarian cysts, and metritis [33]. The vaginal discharge score, otherwise, depends on continuous monitoring at 7 days after birth, using the appropriate technique. However, this management helps in the early detection of metritis, a common infectious disease in dairy cattle [34], allowing for timely intervention to improve cows’ production efficiency once cows with metritis exhibit reduced milk production and fertility and are more likely to be culled [35]. In summary, keeping detailed records of these two specific health issues can help farmers in planning the AI at the most suitable time and choosing bulls that have a higher probability of successful pregnancies.

The THI maximum on the day of AI and the season were the two main environmental variables entering the models. There is a strong influence of the environment on parameters related to animal pregnancy in the context of production, both for beef and dairy cattle. Previous studies [36,37] have identified a low-magnitude heritability for the probability of pregnancy at the first natural mating in Murrah breed heifers (0.122 to 0.154) and the probability of calving from the first insemination in Angus breed heifers (0.025 to 0.048).

In the evaluation of the 26 tested models, the logistic regression model with stepwise feature selection exhibited a notable performance. It secured the top position for the accuracy and negative predictive value, ranked second for the Kappa statistic and sensitivity, fifth for the positive predictive value, and twelfth for the specificity. Logistic regressions are suitable for simple data with linear relationships between variables and outcomes. Moreover, logistic models are easy to implement and explain, efficient with small to moderate datasets, have a low risk of overfitting, and provide a probabilistic output. Consequently, logistic models are constantly used in predicting binary outcomes as is the case of pregnancy probabilities [38,39]. Failure to control the variables entering models according to prior knowledge and in addressing the collinearity between the variables will result in the poor performance of logistic regression models [40]. In the present study, we controlled these aspects; therefore, the simple logistic regression model outperforms some of the machine learning methods. Thurmond et al. [41], employing a Bayesian hierarchical logistic survival model to incorporate longitudinal data from multiple pregnancies of a single cow, revealed that the predicted probabilities of abortion increased with the cow’s age at conception, the number of previous abortions, and if the previous pregnancy was aborted after 60 days of gestation.

The variables that entered the final logistic model were age, parity, retained fetal membranes, the vaginal discharge score, activity at estrus day, the estrus score, reproductive program, AI time after AAM alarm, and bull semen. Mendes et al. [42] showed that cows with moderate production but a lower reproductive performance demonstrated lower longevity in the production system. This issue becomes even more challenging with high-production cows, as milk production shows a negative association with reproductive characteristics [43,44]. Considering these elements, the age of the cow and her physical state play determining roles in the reproductive success of the production system.

On the other side, when employing a pregnancy probability model for semen investment management, the accurate prediction of positive values assumes heightened significance. This is particularly crucial as high-value semen is allocated to cows with a higher likelihood of pregnancy. In such scenarios, the random forest models, utilizing both FULL and GA datasets, outperform other models, resulting in approximately only 10% false positives (Table 2). Moreover, the random forest with the FULL dataset presented the highest Kappa statistic. These models can be applied as robust decision-making tools, allowing for the selection of specific semen for cows predicted to achieve pregnancy, thereby yielding a substantially higher pregnancy ratio. Consequently, this approach has the potential to amplify genetic gain and deliver a more favorable return on investment.

Random forest is a nonlinear tree-based integrated learning model. It is a popular machine learning procedure introduced by Breiman [45]. The forest is composed of many decision trees, and there is no correlation between each decision tree. After the random forest model is obtained, each decision tree in the random forest is judged when the new sample enters. For the classification problem, the voting method is used, and the maximum number of votes is the final model output [40]. Random forests consistently offer the highest prediction accuracy compared to other models in the setting of classification [46]. In the agriculture and livestock sector, several studies have shown good results in the application of random forests, especially for the prediction of binary or class outcomes [47]. For example, random forest methods exhibited a good performance in the prediction of survival to second lactation in Holstein cattle [48], modeling the milk yield of dairy cows under heat stress conditions [49], and also in the prediction of the risk of tick presence on livestock farms [50].

Furthermore, to optimize animal reproduction, it is crucial to associate management strategies that can promote better efficiency in reproductive animal management. In this regard, there are already several studies proposing the use of AAM systems and exploring machine learning algorithms to enhance the prediction of health and fertility disorders in dairy cows [51,52,53]. A previous study has also shown that routinely collected farm data and milk production records on the test day are valuable for predicting the success of insemination in dairy cows [54]. Our preliminary study found that optimizing activity monitoring models by including on-farm measures such as parity, peripartum health history, and environmental conditions can favor the correct identification of estrus and improve activity monitoring alerts regarding the optimal timing for AI, thereby increasing the reproductive performance in dairy cows [11].

Our study integrating AAM systems with specific cow and environmental data to improve pregnancy success in dairy cows showed promise but faces challenges. Issues include the limited generalizability due to the sample diversity, potential model biases, environmental variability, and the economic and technical demands of implementation. Although AAM systems offer advantages over traditional methods and reduced labor, integrating these into less technologically advanced farms is challenging. Ongoing research is necessary to assess the effectiveness and feasibility of these models across different farm settings.

## 5. Conclusions

Including cow-specific variables and environmental factors in machine learning models significantly improves the accuracy, sensitivity, and specificity of predicting pregnancy at the time of artificial insemination in dairy cows using automated activity monitoring (AAM) systems. The most accurate model was the support vector machines (GA) logistic regression, which used AAM data, cow health indicators, and environmental variables. This model showed an accuracy of 87% compared to 84% for the model using only AAM data. These findings can impact decision-making regarding semen investment during AI and highlight the practical relevance and future perspectives of integrating such models in dairy farming. Future studies should explore similar methodologies for embryo recipients to enhance assisted reproduction programs. 

## Figures and Tables

**Figure 1 animals-14-01567-f001:**
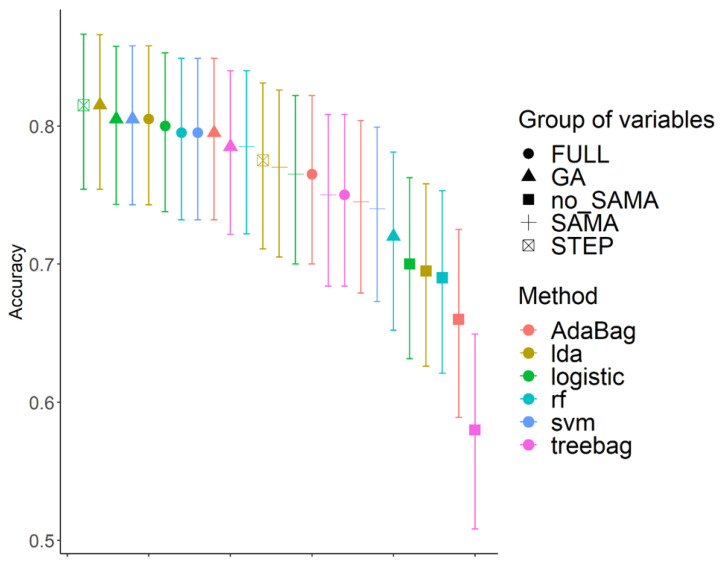
Accuracy of pregnancy prediction for different models using 5 different groups of variables and 6 different methods (showing the average as point and the 95% confidence interval estimation).

**Table 1 animals-14-01567-t001:** Variables included in each variable set used in the models to predict pregnancy 30 days after the first artificial insemination (AI) in dairy cows using an automatic monitoring system (AAM).

Variables Available	Variable Set					
	FULL	No-AAM	AAM	GA	STEP-L	STEP-LDA
Cow-level ^1^						
Age	√	√			√	
Parity	√	√			√	√
Body condition score	√	√				
Locomotion score	√	√				
Retained fetal membranes	√	√		√	√	
Vaginal discharge score	√	√		√	√	√
Peripartum disease	√	√				
DIM on AI day	√	√				
Milk yield on AI day	√	√				
SCC on AI day	√	√				
AAM ^2^						
Activity at estrus day	√		√	√	√	√
Estrus score	√		√	√	√	√
Rumination at estrus day	√		√			√
Breeding ^3^						
Reproductive program	√	√			√	
AI time after AAM alarm	√	√			√	√
Bull	√	√			√	√
Environmental factors ^4^						
Season on AI day	√	√		√	√	
Relative humidity on AI day	√	√				
THI (Max.) on AI day	√	√		√	√	
THI (Min.) on AI day	√	√				√
THI (Mean) on AI day	√	√				

^1^ Parity = primiparous or multiparous; body score condition = 1 (thin) and 5 (fat) according to Ferguson et al. [21]; locomotion score = 1 = walked normally and 5 = presented lameness according to Sprecher et al. [22]; vaginal discharge score collected at 11 ± 4 days postpartum = 1 (clear or translucent), 2 (little purulent material), 3 (mucopurulent), 4 (50% or more pus), and 5 (fetid watery reddish-brown) according to Sheldon et al. [23]; peripartum disease = hypocalcemia, ketosis, displacement abomasum, laminitis, foot disorders, pneumonia, and clinical mastitis; DIM = days in milk; AI = artificial insemination; SCC = somatic cell count. ^2^ Automated activity monitoring (SCR Engineering, Netanya, Israel); estrus score = estrus intensity to predict fertility (0 = no fertility and 100 = high fertility). ^3^ Reproductive program = synchronization of estrus using prostaglandin (PGF 0.5 mg cloprostenol, Sincrocio^®^, Cravinhos, Brazil), or synchronization of estrus and ovulation with estradiol and progesterone protocols [24]. ^4^ Seasons defined for the Southern Hemisphere: spring (21 September to 20 December), summer (2 December to 20 March), fall (21 March to 20 June), and winter (June 21 to September 20).

**Table 2 animals-14-01567-t002:** Performance of machine learning algorithms to predict pregnancy at first postpartum AI in dairy cows using an automated monitoring system ^1^ associated with cow-level ^2^, breeding management ^3^, and environmental factors ^4^.

Model ^5^	Goodness of Fit Measures ^6^							
	Kappa	Se	Sp	PPV	NPV	FP	FN	AUC
no_AAM								
logistic	0.37	0.53	0.80	0.59	0.75	0.41	0.25	0.68
lda	0.31	0.47	0.82	0.60	0.73	0.40	0.27	0.71
rf	0.23	0.28	0.92	0.67	0.69	0.33	0.31	0.68
AdaBag	0.23	0.43	0.79	0.54	0.71	0.47	0.29	0.65
svm	-	-	-	-	-	-	-	-
treebag	0.11	0.47	0.64	0.43	0.68	0.58	0.32	0.57
AAM								
rf	0.50	0.54	0.92	0.80	0.78	0.20	0.22	0.82
lda	0.47	0.56	0.89	0.74	0.78	0.26	0.22	0.82
logistic	0.46	0.56	0.88	0.73	0.78	0.27	0.22	0.84
treebag	0.43	0.56	0.86	0.69	0.77	0.31	0.23	0.81
AdaBag	0.40	0.46	0.91	0.73	0.75	0.27	0.25	0.77
svm	0.37	0.40	0.93	0.76	0.74	0.24	0.27	0.82
FULL								
lda	0.57	0.67	0.88	0.76	**0.83**	0.24	0.18	0.85
logistic	0.55	0.64	0.89	0.77	0.81	0.23	0.19	0.85
rf	**0.81**	0.50	0.96	0.88	0.77	0.12	0.23	0.86
svm	0.54	0.64	0.88	0.75	0.81	0.25	0.19	0.84
AdaBag	0.45	0.50	0.91	0.77	0.77	0.23	0.24	0.82
treebag	0.42	0.51	0.88	0.71	0.76	0.29	0.24	0.80
STEP								
logistic	0.59	**0.68**	0.89	0.78	**0.83**	0.22	**0.17**	0.86
lda	0.49	0.58	0.88	0.74	0.79	0.26	0.21	0.83
GA								
lda	0.59	**0.68**	0.89	0.78	**0.83**	0.22	**0.17**	0.86
Logistic	0.57	**0.68**	0.88	0.75	**0.83**	0.25	**0.17**	**0.87**
Svm	0.57	0.67	0.88	0.76	**0.83**	0.24	0.18	0.86
AdaBag	0.53	0.59	0.91	0.78	0.80	0.22	0.20	0.84
Treebag	0.53	**0.68**	0.84	0.71	0.82	0.29	0.18	0.84
Rf	0.28	0.25	**0.98**	**0.90**	0.70	**0.10**	0.30	0.86

The best values in each parameter are highlighted in bold. ^1^ SCR Engineering, Netanya, Israel, activity at estrus, rumination at estrus, estrus score (0 = no fertility and 100 = high fertility). ^2^ Parity = primiparous or multiparous; body score condition = 1 (thin) and 5 (fat) according to Ferguson et al. [21]; locomotion score = 1 = walked normally and 5 = presented lameness according to Sprecher et al. [22]; vaginal discharge score collected at 11 ± 4 days postpartum = 1 (clear or translucent), 2 (little purulent material), 3 (mucopurulent), 4 (50% or more pus), and 5 (fetid watery reddish-brown) according to Sheldon et al. [23]; peripartum disease = hypocalcemia, ketosis, displacement abomasum, laminitis, foot disorders, pneumonia, and clinical mastitis; DIM = days in milk; AI = artificial insemination; SCC = somatic cell count. ^3^ Reproductive program = synchronization of estrus using prostaglandin (PGF 0.5 mg cloprostenol, Sincrocio^®^, Cravinhos, Brazil), or synchronization of estrus and ovulation with estradiol and progesterone protocols [24], artificial insemination time after AAM alarm (8 h or 10 h), and bull. ^4^ Season on AI day (defined for the Southern Hemisphere: spring = 21 September to 20 December, summer = 2 December to 20 March, fall = 21 March to 20 June, and winter = June 21 to September 20), temperature on AI day, relative humidity on AI day, temperature humidity index on AI day (THI). ^5^ Variables included in the models: no-AAM (variables related to cow-level, breeding management programs, and environmental factors), AAM (only variables used in the system = activity, rumination, and estrus score), FULL (no-AAM and AAM models merged), STEP (selected by stepwise), GA (retained fetal membranes, vaginal discharge at 11 ± 4 DPP, AAM activity at estrus, estrus score, season on AI day, and THI maximum). ^6^ PPV = positive predictive value, NPV = negative predictive value, AUC = area under the curve.

## Data Availability

Data will be made available on request.

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
