# Peer review of "Machine Learning to Predict Pregnancy in Dairy Cows: An Approach Integrating Automated Activity Monitoring and On-Farm Data"

_animals, 2024, doi:10.3390/ani14111567_

Round 1

Reviewer 1 Report

Comments and Suggestions for Authors

Simple summary

What is AAM? Please define it.

Do you mean insemination time with the expression “pregnancy timing”?

Abstract is clear and understandable. In addition, mention all used algorithms. Because some confusing in introduction section (line 64).

Introduction section:

After line 63: Please emphasize the importance of the statistical models for determining the estrus detection time.

Line 123: Please explain why you used Jaccard dissimilarity?

Material and methods are very succesful.

Results

You mentioned the ROC curve in material methods, however I can’t see in the results section. Please add a small paragraph about your summery your results.

Discussion section:

Line 230: Where is ROC curves.

 Conclusion is enough.

Author Response

Thank you for your feedback on our manuscript. We carefully responded to all comments and queries in the manuscript marked in yellow. Additionally, we have revised the entire manuscript, with these changes highlighted in blue. We hope these changes align with the expectations of the reviewers and the editor to be considered further for publication in the Animals.

Answers to the reviewers’ comments and suggestions:

Reviewer 1 answers:

Simple summary

  • What is AAM? Please define it.

ANSWER: Thanks for catching it. We replaced “automated systems that monitor their activities” with “automated activity monitoring systems (AAM) to track their activities.”

  • Do you mean insemination time with the expression “pregnancy timing”?

ANSWER: We appreciate your comment. The expression “pregnancy timing” means “pregnancy at the artificial time”. We rewrote it to be more specific and clearer to the reader. 

Abstract is clear and understandable. In addition, mention all used algorithms. Because some confusing in introduction section (line 64).

ANSWER: Thanks for your comment. We included all algorithms used in our study.

Introduction section:

  • After line 63: Please emphasize the importance of the statistical models for determining the estrus detection time.

ANSWER: Thanks for your suggestion. We included a statement about it to be clear to the reader.

  • Line 123: Please explain why you used Jaccard dissimilarity?

ANSWER: Thanks for your question. The Jaccard index, also known as the Jaccard similarity coefficient, is a statistic used for gauging the similarity and diversity of sample sets. Jaccard similarity is a metric to find similarity between two sets of data, typically used for comparing text data. The dataset about model characteristics has several pieces of information (yes or no) about each model, as, for example, if it is for classification, regression, Bayesian model, boosting, discriminant analysis, kernel method, etc... Therefore, we used Jaccard index to identify the maximum dissimilarity sampling to pick out a diverse set of models. We started using a logistic regression model, therefore we selected, based on these tags, what other four models that constitute the most diverse set.

Material and methods are very successful.

ANSWER: Thanks for your comment.

Results

  • You mentioned the ROC curve in material methods, However I can’t see in the results section.

ANSWER: Thanks for your observation. We wrongly used “ROC curve” instead of “AUC” presented in Table 2. We replaced it in the entire manuscript.

  • Please add a small paragraph about your summary your results.

ANSWER: Thanks for your suggestion and we added a summary of our results.

Discussion section - Line 230: Where is ROC curves.

ANSWER: Thanks for pointing out it. As we mentioned before, we wrongly used “ROC curve” instead of “AUC”. We replaced it in the entire manuscript.

Conclusion is enough.

ANSWER: Thanks for pointing

Reviewer 2 Report

Comments and Suggestions for Authors

Thank you for inviting me to review this manuscript.  The work describes a preliminary study assessing the potential for machine learning to have a role in pregnancy diagnosis on farms and is nicely written.  I just have some minor suggested revisions - see below.  

Main points:

Materials and methods

There is no reference to ethical approval - brief inclusion of this information is needed.  Additionally, there is limited indication of how the animals were diagnosed, and importantly if this was done by the farmer, a veterinary surgeon, or the researchers.  This is important information that warrants inclusion.  There also needs to be some more information on which R package was used for the modelling as this is currently unclear.     

Discussion

There is no discussion of the limitations of this project and the discussion generally is quite brief.  Further discussion of some of the points raised here (for example how the performance of pregnancy diagnosis reported here compares to established pregnancy diagnosis methods would be of interest) and limitations of the study would be beneficial.     

Minor points

Line 45: It would read better here if 'i.e.' (in ‘i.e. body condition score…’) was replaced with 'e.g.' meaning 'for example'.  Currently it reads as if your list of examples is exhaustive, which is not really feasible given the vast number of factors that can affect reproductive performance.

Line 49: It would be beneficial to include a citation here to support your statement that cows in intensive dairy systems show oestrus behaviours for shorter duration to other cows.

Lines 57-63: I find this section a little hard to follow and it is not clear what point you are trying to get across.  Consider re-writing this section for improved clarity.  Inclusion of some supporting citations would be appropriate here also.

Lines 88-89: As mentioned earlier, how were these postpartum diseases defined for the purpose of this study? Additionally, how were they diagnosed, and by who (was this veterinary diagnosis or farm records, which are known to be less accurate)?

Lines 90-95: There seems to be some confusion here between metritis and endometritis.  The Sheldon citation you use (ref no 14) and the scoring system both refer to endometritis, not metritis.  However, these scores are validated for cows > 21 DIM to diagnose endometritis, whereas you have used them to diagnose cows < 21 DIM with metritis – a different condition.  This is a limitation to this study that needs resolving.  I would suggest looking at Sheldon et al. (2006) for further information on differentiating diagnosis of endometritis and metritis. 

Line 101-102: Pereira (ref no 15) describes two progesterone protocols – please clarify which protocol was used in your study. 

Line 237: Starting a paragraph with ‘it’ is a bit vague – I would recommend you considered rephrasing this as ‘the current study highlights….’ Or something similar to indicate that you are referring to your study. 

Lines 289-306: This section at the end of the discussion mentions some important points such as what a random forest model is, but it is written quite briefly with minimal depth and reads as a bit of a list rather than a discussion.  I suggest that this section is expanded a bit more to discuss in some more depth these concepts.       

Table 2: The footnote states that the best values are highlighted in bold but for some parameters a range of values is highlighted.  I think most people would consider that ‘best’ refers to the single most optimal value rather than a range so it might help the reader here if you could define what you mean by ‘best’.    

Suggested reference: Sheldon, I. M., Lewis, G. S., LeBlanc, S., & Gilbert, R. O. (2006). Defining postpartum uterine disease in cattle. Theriogenology, 65(8), 1516–1530. https://doi.org/10.1016/j.theriogenology.2005.08.021

Author Response

Thank you for your feedback on our manuscript. We carefully responded to all comments and queries in the manuscript marked in yellow. Additionally, we have revised the entire manuscript, with these changes highlighted in blue. We hope these changes align with the expectations of the reviewers and the editor to be considered further for publication in the Animals.

Answers to the reviewers’ comments and suggestions:

Reviewer 2 answers:

Thank you for inviting me to review this manuscript.  The work describes a preliminary study assessing the potential for machine learning to have a role in pregnancy diagnosis on farms and is nicely written.  I just have some minor suggested revisions - see below. 

ANSWER: Thank you for pointing out your concerns about our manuscript and acknowledging the merit of our work.

Materials and methods.

  • There is no reference to ethical approval - brief inclusion of this information is needed.

ANSWER: Thanks for your comment. As we used only the data available in the farm software, ethical approval was not needed. However, we included the “Informed Consent Statement” at the end of the manuscript, where the owner of the farm agreement was.

  • Additionally, there is limited indication of how the animals were diagnosed, and importantly if this was done by the farmer, a veterinary surgeon, or the researchers. This is important information that warrants inclusion.

ANSWER: Thanks for pointing it out. As mentioned earlier, we used only the data available in the farm software. However, we included it in the Material and Methods.

  • There also needs to be some more information on which R package was used for the modelling as this is currently unclear.

ANSWER: Thanks for your comment. The “caret package”, cited in the second paragraph on “Building Prediction Models”, was used for all models evaluated.

Discussion. There is no discussion of the limitations of this project and the discussion generally is quite brief.  Further discussion of some of the points raised here (for example how the performance of pregnancy diagnosis reported here compares to established pregnancy diagnosis methods would be of interest) and limitations of the study would be beneficial.    

ANSWER: Thanks for your suggestion. We included it at the end of the discussion.

Line 45: It would read better here if 'i.e.' (in ‘i.e. body condition score…’) was replaced with 'e.g.' meaning 'for example'. Currently it reads as if your list of examples is exhaustive, which is not really feasible given the vast number of factors that can affect reproductive performance.

ANSWER: We appreciate your comment. We rewrote this section for clarity.

Line 49: It would be beneficial to include a citation here to support your statement that cows in intensive dairy systems show oestrus behaviours for shorter duration to other cows.

ANSWER: Thanks for your observation. We added a citation to support our statement.

Lines 57-63: I find this section a little hard to follow and it is not clear what point you are trying to get across.  Consider re-writing this section for improved clarity. Inclusion of some supporting citations would be appropriate here also.

ANSWER: Thanks for pointing out it. We rewrote this section to be more specific and clearer to the reader. 

Lines 88-89: As mentioned earlier, how were these postpartum diseases defined for the purpose of this study? Additionally, how were they diagnosed, and by who (was this veterinary diagnosis or farm records, which are known to be less accurate)?

ANSWER: We appreciate your inquiries. The postpartum diseases examined in our study were derived from the data available in the farm software. Information about the data collection process was included in the manuscript.

Lines 90-95: There seems to be some confusion here between metritis and endometritis. The Sheldon citation you use (ref no 14) and the scoring system both refer to endometritis, not metritis.  However, these scores are validated for cows > 21 DIM to diagnose endometritis, whereas you have used them to diagnose cows < 21 DIM with metritis – a different condition.  This is a limitation to this study that needs resolving.  I would suggest looking at Sheldon et al. (2006) for further information on differentiating diagnosis of endometritis and metritis.

ANSWER: Thank you for pointing it out. We incorrectly listed the reference. We used Sheldon et al. (2006), as mentioned in the Table 1 footnote. We have updated the References section to reflect the correct publication.

Line 101-102: Pereira (ref no 15) describes two progesterone protocols – please clarify which protocol was used in your study.

ANSWER: Thanks for your comment. We added the protocol in the Material and Methods to be clearer to the reader.

Line 237: Starting a paragraph with ‘it’ is a bit vague – I would recommend you considered rephrasing this as ‘the current study highlights….’ Or something similar to indicate that you are referring to your study.

ANSWER: Thanks for your comment. We accepted your suggestion and changed the sentence.

Lines 289-306: This section at the end of the discussion mentions some important points such as what a random forest model is, but it is written quite briefly with minimal depth and reads as a bit of a list rather than a discussion.  I suggest that this section is expanded a bit more to discuss in some more depth these concepts.      

ANSWER: We appreciate your comment. We rewrote it to be more specific and clearer to the reader. 

Table 2: The footnote states that the best values are highlighted in bold but for some parameters a range of values is highlighted.  I think most people would consider that ‘best’ refers to the single most optimal value rather than a range so it might help the reader here if you could define what you mean by ‘best’.   

ANSWER: Thanks for your comment. We highlighted only the best values for each variable.

Suggested reference: Sheldon, I. M., Lewis, G. S., LeBlanc, S., & Gilbert, R. O. (2006). Defining postpartum uterine disease in cattle. Theriogenology, 65(8), 1516–1530. https://doi.org/10.1016/j.theriogenology.2005.08.021

ANSWER: We appreciate for catching it. We incorrectly listed the reference. We used Sheldon et al. (2006), as mentioned in the Table 1 footnote. We have updated the References section to reflect the correct publication.

Reviewer 3 Report

Comments and Suggestions for Authors

- The article deals with current issues, important for farms with intensive breeding of dairy cattle. I have several questions and comments about the article.

- The presented article is based on the results of research presented by some of the authors: Artificial insemination timing on pregnancy rate of Holstein cows using an automated activity monitoring. Ciência Rural, Santa Maria, v.54:03, e20220557, 2024. https://doi.org/10.1590/0103-8478cr20220557  

- However, in References there is a citation under number 9, as if this publication was published in 2023:

References 9. Marques, L.R.; Almeida, J.V.N.; Oliveira, A.C.; Paim, T.P.; Marques, T.C.; Leão, K.M. Artificial insemination timing on pregnancy rate of Holstein cows using an automated activity monitoring. Cienc Rural 2023, 54:e20220557, doi:10.1590/0103-8478cr20220557.“

The authors should explain and possibly correct this difference.

- It would be appropriate to state what is the benefit of the so-called "Machine learning...", which is highlighted in the title of this new article, compared to the previous article.

- How do the authors account for farm size and herd size? The research was conducted only on one farm with a considerably large capacity of reared dairy cows.

- The article does not mention how the dairy cows were housed, what the cowshed was like, and which type the milking equipment was there. This can be important from the point of view of the microclimate. What were the methods and sensors used for identification?

- The authors verified their method on the classical dairy breed Holstein cows. How do they consider the influence of the breed of dairy cows? Can this method also be used for other breeds, e.g. Jersey cows? What could be the difference?

- The authors state that they also take into account the influence of the environment. How and for which different kinds of environments did they validate their model? According to the aforementioned article from 2024, the environment during the 4 seasons was: "Temperature (°C): Spring 25.3±0.2, Summer 24.7±0.2, Fall 22.5±0.1, Winter 23.1±0.2" which are very balanced temperatures, compared to climatic conditions of many countries of the world, where the differences between temperatures, for example, in winter (-15°C) and summer (+35°C) are very significant. It would be very interesting to verify the investigated model even in such different climatic conditions.

- The article states that the productivity of dairy cows was year-round (kg of milk/cow·year), and how the influence of the environment, e.g. season, etc., manifested itself.

In the article, it would be appropriate to add a comparison and the benefit of how much better the newly developed model is compared to the existing procedures and estimates of pregnancy prediction in dairy cows.

- Conclusions – this final part is very general. It is necessary to supplement with specific results and to emphasize the contribution of the conducted research in terms of real values.

Author Response

Thank you for your feedback on our manuscript. We carefully responded to all comments and queries in the manuscript marked in blue. Additionally, we have revised the entire manuscript, with these changes highlighted in yellow. We hope these changes align with the expectations of the reviewers and the editor to be considered further for publication in the Animals.

Answers to the reviewers’ comments and suggestions:

Reviewer 3 answers:

The article deals with current issues, important for farms with intensive breeding of dairy cattle. I have several questions and comments about the article.

ANSWER: We appreciate your time and concerns about our manuscript. We made changes to improve our work.

The presented article is based on the results of research presented by some of the authors: Artificial insemination timing on pregnancy rate of Holstein cows using an automated activity monitoring. Ciência Rural, Santa Maria, v.54:03, e20220557, 2024. https://doi.org/10.1590/0103-8478cr20220557. However, in References there is a citation under number 9, as if this publication was published in 2023: References 9. Marques, L.R.; Almeida, J.V.N.; Oliveira, A.C.; Paim, T.P.; Marques, T.C.; Leão, K.M. Artificial insemination timing on pregnancy rate of Holstein cows using an automated activity monitoring. Cienc Rural 2023, 54:e20220557, doi:10.1590/0103-8478cr20220557. The authors should explain and possibly correct this difference.

ANSWER: Thank you for pointing it out. We incorrectly listed the year as 2023. We have updated the References section to reflect the correct publication year, 2024.

It would be appropriate to state what is the benefit of the so-called "Machine learning...", which is highlighted in the title of this new article, compared to the previous article.

ANSWER: Thanks for your suggestion. We included your suggestion.

How do the authors account for farm size and herd size? The research was conducted only on one farm with a considerably large capacity of reared dairy cows.

ANSWER: Thanks for your question. The farm has 300 lactating cows, and the research was carried out with data referring to 3 years (2018 to 2020).

The article does not mention how the dairy cows were housed, what the cowshed was like, and which type the milking equipment was there. This can be important from the point of view of the microclimate. What were the methods and sensors used for identification?

ANSWER: Thank you for highlighting it. As noted in the Materials and Methods section, the methodology is detailed in our previously published manuscript (Marques et al., 2024). To avoid plagiarism, we decided not to repeat these aspects in the current document. However, due to your suggestion, we briefly described these aspects.

The authors verified their method on the classical dairy breed Holstein cows. How do they consider the influence of the breed of dairy cows? Can this method also be used for other breeds, e.g. Jersey cows? What could be the difference?

ANSWER: Thanks for your question. The method verified on Holstein cows may need adjustments to apply to other breeds like Jersey cows, due to genetic, physiological, and behavioral differences between breeds. These variations can influence traits such as milk composition and fertility rates, potentially affecting model accuracy. Adapting this method for other breeds would likely require breed-specific data collection and model retraining or fine-tuning to address these disparities and ensure effective predictions across different dairy breeds.

The authors state that they also take into account the influence of the environment. How and for which different kinds of environments did they validate their model? According to the aforementioned article from 2024, the environment during the 4 seasons was: "Temperature (°C): Spring 25.3±0.2, Summer 24.7±0.2, Fall 22.5±0.1, Winter 23.1±0.2" which are very balanced temperatures, compared to climatic conditions of many countries of the world, where the differences between temperatures, for example, in winter (-15°C) and summer (+35°C) are very significant. It would be very interesting to verify the investigated model even in such different climatic conditions.

ANSWER: Thanks for your question. Thinking on environmental conditions, the current validation of the model under these balanced conditions could apply to tropical regions, situated between the Tropic of Cancer and the Tropic of Capricorn, which are characterized by high humidity and significant annual rainfall, typically divided into pronounced wet and dry seasons. It suggests it may not be fully applicable or accurate in regions with more extreme and variable climates. To improve the global robustness and applicability of the model, it is important to test and possibly adjust it in various environments with significant temperature variations. This approach would provide insights into how extreme temperatures affect the model's ability to predict cow fertility accurately, potentially leading to modifications that would enhance its effectiveness in diverse climatic conditions.

The article states that the productivity of dairy cows was year-round (kg of milk/cow·year), and how the influence of the environment, e.g. season, etc., manifested itself. In the article, it would be appropriate to add a comparison and the benefit of how much better the newly developed model is compared to the existing procedures and estimates of pregnancy prediction in dairy cows.

ANSWER: We appreciate your suggestion. It is not used routinely any estimation of pregnancy probability at the moment of artificial insemination, which we are propounding in this paper. The farms currently using the automatic monitoring system (AAM) use the "estrus score" as an indication of pregnancy probability. However, we showed that using other parameters would better estimate this probability.

Conclusions – this final part is very general. It is necessary to supplement with specific results and to emphasize the contribution of the conducted research in terms of real values.

ANSWER: Thank you for pointing out your concerns about our manuscript. We rewrote the conclusion to cover your sug

Round 2

Reviewer 3 Report

Comments and Suggestions for Authors

The article was partially edited according to the reviewer's instructions and the authors' possibilities. I recommend it for publication.